# Retrograde Free Venous Flaps for Extremity Reconstruction: A Roadmap

**DOI:** 10.3390/medicina58081065

**Published:** 2022-08-07

**Authors:** Thomas Giesen, Olga Politikou, Ivan Tami, Maurizio Calcagni

**Affiliations:** 1Centro Manoegomito, Clinica Ars Medica, 6900 Gravesano, Switzerland; 2Faculty of Biomedical Sciences, Università della Svizzera Italiana, 6900 Lugano, Switzerland; 3Department of Plastic Surgery and Hand Surgery, University Hospital Zurich, 8091 Zurich, Switzerland

**Keywords:** hand reconstruction, extremities reconstruction, reconstructive microsurgery, flaps, free flaps, free venous flaps, venous flaps, composite flaps

## Abstract

*Background and Objectives*: Retrograde free venous flaps represent a separate entity among free venous flaps: their physiology is still unclear, but they provide an immediate visible refill after reconnection, with a similar behaviour to conventional flaps. Therefore, the dimensions and the indications of these flaps can be extended beyond what was previously believed, and they can be easily customized, including with respect to tendons and nerves. Nevertheless, they are still debated and regarded as unsafe. *Materials and Methods*: From 2012 to 2019, we performed 31 retrograde free venous flaps on 31 patients to reconstruct hands, digits, and in one case the heel. All the flaps were arterialized in a retrograde manner; the donor site was the forearm in 28 cases, the foot in 2 cases, and the calf in 1 case. We recorded the size, vein architecture, donor site, donor artery, donor morbidity, function for composite and non-composite flaps, immediate complications, late complications, survival rate, and the number of revisions. We recorded the hand function when appropriate. A total of 10 flaps were also intraoperatively studied with indocyanine green to monitor their hemodynamical behaviour. *Results*: All the patients were followed for an average of 8 months (6–15). The flap dimensions ranged from 6 cm^2^ to 136 cm^2^. All the flaps, except two that had complete necrosis, survived. Two flaps had partial necrosis. There was no correlation between necrosis and the size of the flap, with one case of necrosis and one of partial necrosis in the small flaps (<10 cm^2^). None of the cases with partial necrosis needed a new flap. Two flaps developed a late arterio-venous shunt that was ligated. *Conclusions*: The retrograde free venous flaps proved to be a useful tool for complex reconstructions of the hand and extremities. They can provide a large island of pliable skin and composite tissue with tendons and nerves, but surgeons must be aware of some caveats.

## 1. Introduction

Venous flaps and free venous flaps use a vein as an afferent vessel rather than an artery. The blood drainage is obtained by the same input vein (flowing through the flaps) or by a different vein. These particular flaps have long been known and are mainly used in specific clinical situations such as difficult replantations as a source of acute coverage and as vessel carriers [1,2]. The dimensions of these flaps are normally reduced and they are generally considered unsafe and difficult to monitor as they tend not to show a clinical refill [3]. In general, venous flaps and free venous flaps have long been debated but it has been demonstrated that they can achieve physiological revascularization [4]. Nevertheless, venous flaps are appealing for the simplicity of the harvesting technique, the abundance of donor sites, and the possibility for harvesting these cutaneous flaps as composite flaps including tendons and nerves [2,5,6]. Retrograde free venous flaps represent a separate entity among free venous flaps: in these flaps, the input artery is anastomosed to the designated input vein of the flap in a retrograde manner, i.e., against the valves (Figure 1a). This specific configuration has been shown to be crucial to the behaviour of these flaps, as flaps with this design show an intraoperative visible refill as the incoming blood, once hitting the first valve in the flap, is pushed to the periphery of the flap itself [6,7,8,9] (Figure 1).

We present our experience with 31 retrograde free venous flaps, including composite flaps, used for the reconstruction of the extremities, with a focus on the surgical technique. We report an extremely high survival rate in medium and large flaps with almost no difficulties in healing. We demonstrate how these flaps can be routinely used in elective situations and report that the survival of these flaps is much larger than was previously believed. Upon reporting our results, we highlight and analyse our cases with their complications in order to contribute to the further understanding of these flaps.

## 2. Materials and Methods

The Cantonal Ethics Committee of Zurich, Zurich, Switzerland, provided approval (BASEC-Nr. 2017-02001) for this study.

Between January 2012 and December 2020, 31 patients underwent reconstruction of soft tissue defects of the hand or the lower extremities with retrograde arterialized free venous flaps. The patients included 24 men and 7 women, with a mean age of 41 years (range 17 to 47). A total of 10 patients were overweight (body mass index between 25 and 30). There was 1 patient who was obese. A total of 6 patients were smokers. Diabetes and poor patient compliance were the only exclusion criteria.

### 2.1. Indications and Flap Size

All patients required soft tissue reconstruction to the hand or the distal forearm, except 1 who had an acute soft tissue defect to the ankle. A total of 14 patients had an acute trauma requiring early soft tissue reconstruction. A total of 4 patients were treated acutely for severe infections. The other 13 patients had an elective indication for soft tissue reconstruction. The mean surface area of the flaps was 40 cm^2^. A total of 16 flaps were small (less than 20 cm^2^), 9 flaps were medium-sized (24 to 70 cm^2^), and 6 were large (range 77 to 136 cm^2^). Details of the patients and defects are summarized in Table 1.

### 2.2. Surgical Technique

All patients underwent surgery under general anaesthesia with tourniquet control.

(1)Preparation of the recipient site

In traumatic cases and infections, soft tissues were debrided before harvesting the flap. In extensor teno-arthrolysis, the tenolysis and joint procedures were performed firstly by simply incising the skin. Once this was completed, the affected digit was passively flexed fully, and a template of the missing skin was taken. Then, the feeding artery and the draining veins were prepared for skin reconstruction. A total of 9 flaps were anastomosed to a side branch of the dorsal radial artery and 2 flaps end-to-side to the main dorsal radial artery. A total of 1 flap was anastomosed to the ulnar artery, 5 flaps to a common digital artery, and 5 to a digital artery in the injured finger. The arterial feeding for 8 flaps was sourced from the digital artery harvested from the adjacent digit. One flap was anastomosed end-to-side to the tibial posterior (TP) artery. Dorsal draining veins were used in all cases but 3, where the palmar vein system was used.

(2)Preparation and harvesting of the flap

All visible veins were marked before draping the limb. In all cases, a template of the defect was made, and the defect drawn on the flexor aspect of the forearm in its distal third for small defects, on the middle third for medium defects, and on the proximal third for large defects. In 2 cases, glabrous skin from the thenar and hypothenar eminence was included to restore finger pulp. In 2 cases, the flaps were harvested from the foot and in 1 case from the calf. In 3 cases, the contralateral forearm of the injured hand was used as a donor site due to impeding local reasons: in 1 case there was an infection; in 1 case a large tattoo was present, which the patient did not want altered; and in 1 case there was a compromise of the soft tissue in the same forearm. The shape of the flap was always elliptical and oriented obliquely to the donor limb main axis. Such planning of the flap enables the inclusion of more veins and aids primary closure of the donor site. The position of the flap was also influenced by the necessity of harvesting a composite flap including tendons and/or nerves. When possible, 3 or more veins were included, and the most central vein intersecting the flap was selected to become the arterialized feeding vessel in medium and large flaps (Figure 2). Placement of the flap was influenced by the presence of hair, and in some cases, the need to include cutaneous nerves of the forearm and/or tendons in the flap. In this series, 12 flaps were composite flaps with 6 flaps including a tendon and 6 flaps including a nerve.

(3)Flap transfer and connection to the hand

When the same forearm was used for the flap donor site, the connecting veins of the flap were divided, and the flap was transferred to the hand without releasing the tourniquet. Haemostasis was then achieved after releasing the tourniquet. End-to-end anastomosis of the feeding and draining vessels was carried out with 10/0 nylon under the microscope for all flaps, except 3, where an end-to-side anastomosis to a main limb artery was performed: 1 tibia posterior artery and 2 radial arteries. The flap vein chosen as feeding vessel was always anastomosed to the feeding artery first, and then the clamps were released to facilitate immediate filling of the vascular tree of the flap. Anastomosis of the draining veins of the flap was then performed. In most cases, these veins were still empty when clamped before the anastomosis. Whenever possible, 2 draining veins were anastomosed for large flaps. After completion of the microsurgery, skin suture was completed. Either 1, 2, or 3 simple capillary drains were placed according to the dimensions of the flap.

A total of 10 patients were injected with indocyanine green intraoperatively (ICG) at 5 mg/mL (ICG Pulsion©, Pulsion Medical Systems, Munich, Germany) and the vascular tree of the flap was observed on the table—this occurred immediately after completing the arterial and venous anastomoses. Video recording with an infrared camera of the vascular tree pattern (Kodak, Rochester, NY, USA) was performed with the flap not yet sutured in place and “upside-down”. The patients were injected again with an additional 1 mL (5 mg) ICG intravenously after the final in-setting to record good flap vascularization.

In all cases, the time between arterial anastomosis and visible refill was recorded. The harvesting time was also recorded. All extra vascular procedures conducted in order to obtain visible refill and good vein drainage were recorded.

### 2.3. Postoperative Management

#### 2.3.1. Immediate Postoperative Management

The hand, forearm, and elbow were wrapped in a bulky dressing and were elevated for 5 days while the patients remained hospitalized. Mobilization for personal needs was allowed after 72 h. All patients postoperatively received 10,000 u of heparin IV per 24 h for 4 days.

Flap colour, turgor, bleeding, swelling, and refill time were monitored every two hours by a trained nurse and every 4 h by a surgeon for the first 72 h. Any blistering or epidermolysis of the flaps was also recorded. Monitoring by the nurse continued at 3 hourly intervals for the next 2 days.

The first dressing change occurred 5- or 6-days following the operation (range 5 to 7), and compression bandaging and hand therapy was started at 9 days after surgery (range 7 to 12), unless any necrosis was seen. Hand therapy and flap training occurred with the limb at normal level for incremental periods and with the patient performing intermittent circular movements of the whole upper limb in order to change the blood pressure within the flap.

#### 2.3.2. Long-Term Postoperative Management

All flaps were followed clinically for an average of 8 months (range 6 to 15). One flap was studied with indocyanine green at the 6-month follow up.

##### Assessment

Intra-operative and immediate postoperative assessment included parameters noted in Table 2.

Late postoperative assessment included parameters noted in Table 3.

## 3. Results

Survival rate and functional results are summarized in Table 2 and Table 3.

### 3.1. Survival, Revision, Early and Late Complications

Out of the 31 flaps, 29 survived with no need for further resurfacing procedures (93.5%).

Out of the 31 flaps, 4 flaps needed revision: 2 eventually survived and 2 failed. These four flaps dimensions were variable and comprised three small flaps and one medium flap.

The 2 flaps that were revised but eventually survived were one small (6 cm^2^) and one of medium dimensions (40 cm^2^). The single medium-flap had an early venous congestion and during revision we observed an A-V shunt with a patent draining vein. We ligated the A-V Shunt and then observed a good—not pulsating—flow through the draining vein. The flap had only one vein available for drainage and we did not perform any further procedure.

In the other case, which we revised and in which the flap eventually survived, we found a thrombosis at the level of the draining vein anastomosis. We performed a new anastomosis with an alternative dorsal recipient vein, with success. The flap was small (6 cm^2^) and there were no other veins available for drainage.

In both these two cases, we postoperatively observed a progressive complete superficial epidermolysis in the first few days. The epidermolysis progressed to superficial skin necrosis in the following weeks, but the fat tissue underneath the flap was clearly viable and survived. Both flaps re-epithelized spontaneously with a good aesthetic and functional result.

Of the two flaps that failed, both had an early venous congestion and were revised within 3 h from primary surgery. In both cases, the draining vein anastomosis was patent, but an A-V Shunt was observed with the draining vein displaying a pulsating arterial flow. In both cases, the A-V shunt was not ligated; however, we performed a new anastomosis of the same draining vein of the flap to a new dorsal and larger receiving vein. The congestion did not resolve, and as both flaps were of small dimensions, there were no other veins available for drainage. They both eventually failed, and we performed a second flap for resurfacing.

Of the 29 flaps that survived, 2 flaps had a partial marginal necrosis. In one case, which represents the largest flap of this series (136 cm^2^), we recorded a 15% superficial necrosis that eventually healed by secondary intention. The ischemic area of the flap was at the very distal end of the flap in a patient that had a base systolic pressure of 90 mm Hg. In the second case, we had a marginal necrosis of ca 5% that healed spontaneously (Figure 3). This was a large flap of 80 cm^2^.

In one case, the flap suddenly no longer showed visible refill at ca. 6 h postoperatively: at the bedside of the patient, a small hematoma was suspected, and three sutures were released. The hematoma was gently drained, and the small wound left open. The flap survived with no further incidents and the little opening healed by secondary intention.

In one elective case of the first web space, we had a postoperative infection that healed with antibiotics and no further procedures.

In two cases, we observed a late A-V shunt when the flap was already healed and the function was restored. In these two cases, with the pulsatile vein just under the skin paddle, we decided to ligate the A-V shunt to avoid major bleeding in case of future superficial injuries. The ligature performed 6 months from primary surgery did not alter the flap.

### 3.2. Harvesting Time and Time to Visible Refill

We recorded a harvesting time for this series with an average of 43 min (range 34–60). The flap that took the longest was the large flap harvested from the calf, as it included two vein systems (superficial and deep) of different calibres.

The time between arterial anastomosis and visible refill was an average of 7 min, from a range of 1–65. The median value was 1 min, showing a tendency for an early visible refill. The flap that showed a visible refill with 60 min was the one on the calf.

### 3.3. Intraoperative Extra Microvascular Procedures

In two cases, a valve in the feeding anastomosed vein was observed blocking the input arterial flow before entering the fat and skin paddle. This resulted in only part of the flap showing a slow visible refill. In one case, partial and insufficient vascularization was confirmed by the ICG. In these two cases, the vein chosen as a feeding vein was swept with the draining vein and two new anastomoses were performed. We then observed a full visible refill all over both flaps, and they survived without any further complications. In three cases, an immediate A-V shunt was observed intraoperatively after the venous anastomosis. In two cases, the A-V shunt was immediately ligated: we then observed a good, but not pulsating, flow through the draining vein. In the other case, we anastomosed a second vein with success and with no shunt. All flaps survived with no further complications.

### 3.4. ICG Results

A total of 10 flaps were studied intraoperatively with ICG, firstly “upside-down” (Figure 1b) to visualize the immediate vascular tree and the optimal vascularization after the in-setting. In eight flaps, we observed the signal of the input arterial blood entering the arterialized vein through the first valve. At this point the signal was disappearing, but it reappeared shortly after at the dermal margins of the flap. These eight flaps showed a clinically visible refill on inspection after the operating room lights were switched back on. This behaviour might suggest the presence of a small artero-venous shunt between the venous and arterial system, probably at one of the two vascular plexuses: the deeper at the border between the subdermal fat and dermal layer, or the dermal plexus.

One flap showed peripheral revascularization at the dermal layer, but also a clear A-V shunt within the flap between the arterialized vein and the draining vein. This was a small flap, and no other veins were available in the flap for anastomosis. It was decided to reperform the draining vein anastomoses with a larger dorsal vein. The ICG behaviour did not alter. The flap eventaully failed.

One flap showed a very slow flow of the aterialized vein before entering the flap and an incomplete signal of the dermal layer. With the lights switched on, the flap showed a partially visible refill only in the part of the flap close to the entry point of the arterialized vein. We decided to switch the draining vein to the feeding artery, and we used a third vein as a draining vein. The ICG showed a good revascularization and survived with no further problems.

### 3.5. Donor Site Morbidity

Four flaps were closed partially using a skin graft. All of them were large flaps (>70 cm^2^). Two patients (case 13 and case 21) were not satisfied with the appearance of the donor site and one of them asked for a surgical correction while the other one (case 21) did not want any correction. One patient (case 22), with a relatively large flap that was harvested from the contralateral side, was not satisfied about the appearance of the donor site after direct closure because of a mild dog ear. This patient asked for surgical correction. None of the patients that had a composite flap including cutaneous nerves of the forearm developed neuropathic pain nor complained about the limited impaired sensation in the affected part of the forearm. None of the patients that had a composite flap including tendons complained about impaired postoperative hand or forearm function.

## 4. Discussion

Venous flaps and free venous flaps have long been known and debated. The main concern has always been the amount of unpredictability regarding the survival and difficulties in monitoring [10]. The literature is also not univocal, with several papers publishing good results and viability for these flaps but failing to explain their physiology in a satisfactory manner [10,11,12]. On top of this, in the last few years the hype has been mainly about the superficial circumflex inferior perforator (SCIP) flap. The SCIP flap is a very versatile flap, relatively easy to harvest, and one that can offer a pliable thin skin with minor donor site morbidity and the possibility of including a vascularized bone graft [13]. Nevertheless, retrograde free venous flaps still offer several advantages, such as the following:(1)They are easy and fast to harvest [14], with the possibility of easily including tendons and nerves for complex reconstructions. Differently from other configurations of free venous flaps, the input and output vessels are on the same side of the flap, crucially assisting the surgeon in the in-setting of the flap and in the set-up of the vessels’ configuration.(2)They offer pliable skin, especially when harvested from the forearm, the calf, or the dorsum of the foot. In case of patients with a very high BMI, they can be harvested from the foot as pliable and thin flaps that can also be composite. The need for pliable skin seems to be relevant in a Caucasian population.(3)They can be harvested from potentially every part of the body as counterpart flaps of their conventional version, making them easily available in specific situations in which a proximity-free tissue transfer is needed.

In retrograde arterialized free venous flaps, the input arterial flow is pushed by the closed valves to the periphery of the flap. This mechanism produces an interesting, and until now inexplicable, phenomenon, i.e., the visible revascularization of these flaps with a visible refill. (Figure 2)

We report in our series a relatively high rate of survival, and the success of the large retrograde venous flaps. We have summarized our finding in the following points, hoping to contribute to the understanding, and use, of these specific flaps.

**Survival**: Our rate of survival is 93%. We lost two flaps that both showed venous congestion, which was due to an A-V shunt that in these two cases was not ligated. In both these flaps, we re-anastomosed the draining vein to a new recipient vein. This solution has been shown to be inadequate. In all the other flaps that showed an A-V shunt that was ligated, or when we anastomosed an extra draining vein, the flaps survived. Arguably, if we could have performed an adequate ligature of the A-V shunt in the two flaps we lost, we could have potentially had an even higher survival rate. As already reported [15], the ligature of an immediate A-V shunt seems to be essential for these flaps to survive. As the difference between a proper and an unwanted A-V shunt and a fine vessel’s connection between the input and output system might be difficult to recognize intraoperatively, in the case of a true A-V shunt, we noticed that (a) the flap becomes rapidly congested, with a visible refill under 1 s; (b) the draining vein is strongly pulsating in a synchronized manner with the arterialized vein; and (c) by leaving the vessels connected and flipping the flap upside down, it is very easy to spot under magnification the A-V shunt.

**Design**: Analysing the different flap designs of our series, we established the following set of guidelines for the design of retrograde arterialized free venous flaps. The flap dimension should be slightly larger than the defect: the flaps tend to swell after surgery, regardless of their good venous drainage. An oblique design to the main axis of the donor limb is preferable for a better direct closure and at the same time allowing for the possibility of including more veins. The best possible configuration that helps direct skin closure is having the main axis of the flap oriented from proximal/ulnar to distal/radial. Preferably, the arterialized vein should be at the centre of the flap, but this is not mandatory. If the vein chosen to be arterialized has some side branch before entering the flap, beware, as it is most likely that a valve outside the flap will be present at that point, compromising the good vascularity of the flap. In case the flap does not include many veins, a connecting branch between the feeding arterialized vein and the draining vein should always been checked for an immediate A-V shunt and—as already suggested in the literature—ligated. In our experience, it is preferable to anastomose two draining veins, if possible.

**Monitoring and Revision**: The clinical monitoring process is identical to conventional free flaps: the clinical disappearance of the refill is correlated with a stop in the input flow. A congestion of the flap with swelling, darkening of the skin with darker spots at the margins, profuse marginal bleeding, and hypothermia are signs of a venous problem due to a thrombosis of the draining vein or to a high flow A-V shunt within the flap. These are indications for immediate revision, exactly as in conventional fascio cutaneous flaps. As obvious as this might sound, care should be taken not to judge a clinical venous congestion of a retrograde arterialized free venous flap as para-physiological: these flaps’ hemodynamic nature is very different from the more conventional anterograde arterialized venous flaps, and they closely reproduce the aspect and the behaviour of a conventional fascio cutaneous free flap. As a difference to conventional fascio cutaneous flaps, different refill speeds have been observed, from 1.5 to 3 s, with different refill speeds within large flaps. The refill speed per se did not seem to be a relevant prognostic sign for predicting congestion, but rather the change of refill speed over time. In Figure 3, we summarized our suggestions in case of insufficient blood input or in case of a A-V Shunt.

**Dimension**: In the literature, free venous flaps are considered small if their surface is of 10 cm^2^ and large if their surface is 25 cm^2^ [2]. We slightly changed this definition and in our series we considered only nine flaps as large (77–136 cm^2^). If we were utilizing the classification used by Woo, our series would actually include 15 large flaps. This finding is in our opinion, of paramount importance, as most of the literature is focused on small venous flaps, with few examples of flaps of greater dimension [7]. In our series, none of the large flaps failed, and to us, larger flaps are safer than small flaps. This convinced us to use them in elective cases as well, and in our series of 31 flaps, 13 had an elective indication (Figure 4).

**Physiology:** The vascular supply to the venous flaps’ skin island has been reported in the published literature as coming mainly from the dermal plexus [16]. Nevertheless, the exact physiological behaviour of the retrograde free venous flaps has not yet been clearly demonstrated [4,8,11]. Several elements are worth consideration: recent studies have shown that the ligature of a vein in an anterograde free venous flap, where the input blood is running in favour of the valves, reproduces the same effect as in retrograde free venous flaps: a visible clinical refill [11]. In our flaps studied with ICG, we observed that the fluorescence of the incoming arterial blood was “disappearing” from the arterialized vein at the level of the first venous valve, and then “re-appearing” at the dermal layer from bleeding dermal spots. The reappearance of the fluorescence at the dermal layer was associated with the visible refill of the skin paddle of the flap. These two factors suggest that at the level of the subdermal or the dermal plexus, there is a possible switch of the incoming arterial blood from the vein system to the arterial system. Simplicity would suggest that this should happen at the level of the first encountered plexus, i.e., the deep plexus between the fat tissue and dermal layer. This interpretation is corroborated by two further elements we recorded: (1) The fastest refill time close to the entering point of the arterialized vein in large flaps suggests that close to that area, the venous draining system is somehow “congested”. To us, the simplest answer would be that close to the arterialized vein entering point the drainage is running through only one of the two systems, most likely the small veins of the dermal plexus. (2) In two flaps we revised because of venous congestion, we observed afterwards a progressive epidermolysis that progressed to skin necrosis, while the whole subcutaneous tissue of the flap survived. The two patients did not need any further operation as the defect re-epithelialized spontaneously. This differing behaviour between the dermal and subdermal layer suggests a dualism of hemodynamic function in these flaps.

**Indications:** Our indications ranged from acute traumatic, to acute infection, and to elective scar contractures and 1st web contractures. Arguably, the most interesting and novel indications were the elective ones, i.e., the possibility of performing large 1st web reconstruction with sensate skin, the possibility of performing dorsal teno-arthrolysis of the digits with pliable and thin skin, and the possibility of including tendons at ease.

## 5. Conclusions

Arterialized retrograde free venous flaps have been shown to behave differently from other configurations of venous flaps and to be safe. They are a useful tool in the hand of the microsurgeon and can be considered as a primary tool for reconstructing complex composite defects in the extremities. Nevertheless, their physiology is still unclear and further research is needed to help further understand their behaviour

## Figures and Tables

**Figure 1 medicina-58-01065-f001:**
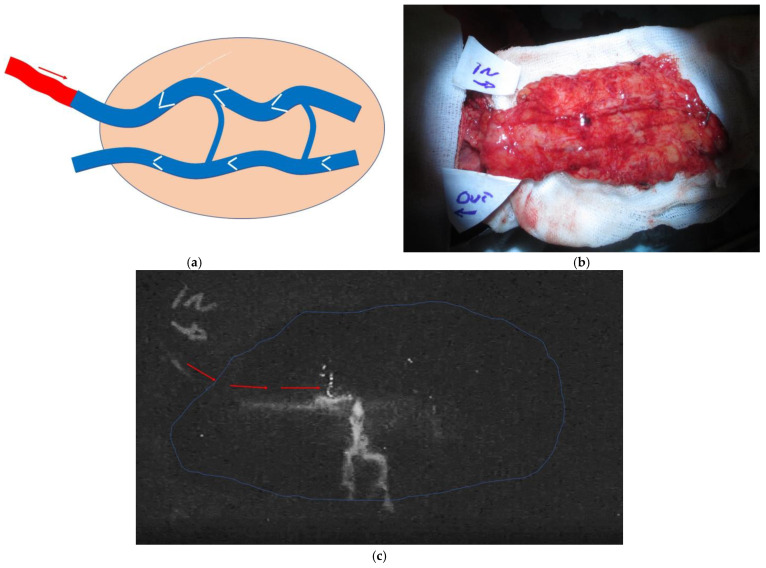
(**a**) In the retrograde free venous flaps, the input arterial flow (red arrow) is entering the flap through a vein against the valve; (**b**) In this case, more detailed in Figure 2, the flap was studied upside down with ICG green to document the revascularization pattern once the artery and the draining vein were sutured; (**c**) the same flap at time 0, when the ICG started flowing into the arterialized vein; (**d**) The flap after 5 min with diffuse revascularization. The strong stream signal in the centre is a leak from the arterialized vein.

**Figure 2 medicina-58-01065-f002:**
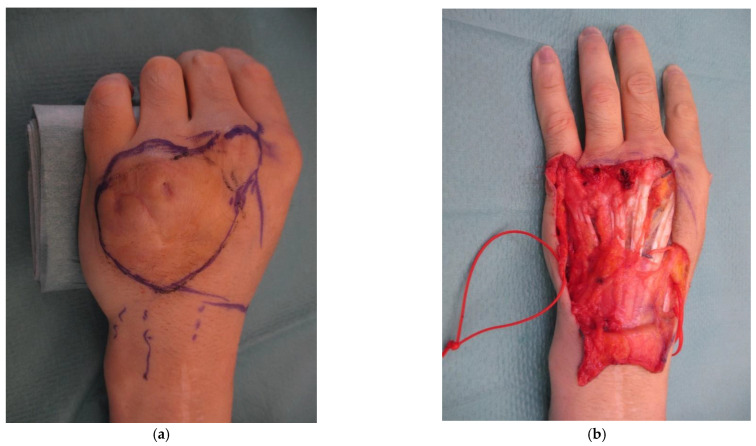
(**a**) 40-year-old male patient with a full thickness chemical burn to the dorsum of the non-dominant left hand and extensor tendon adhesions; (**b**) Debridement of the burned skin and tenolysis with a resulting defect of 30 cm^2^ (6 cm × 5 cm); (**c**) retrograde free venous flap harvested from the contralateral forearm because of large tattoo on the left forearm. Red arrow indicates the vein chosen as the one to be arterialized. Blue arrows indicate the draining vein; (**d**) flap in place, anastomosed end-to end to a collateral branch of the dorsal radial artery and (**e**) immediate visible refill (ca. 5 min); (**f**) situation at 12 days, with no blistering, excessive swelling, or discolouring; (**g**) results at 6 months with full finger extension; (**h**) full flexion of the finger was also achieved.

**Figure 3 medicina-58-01065-f003:**
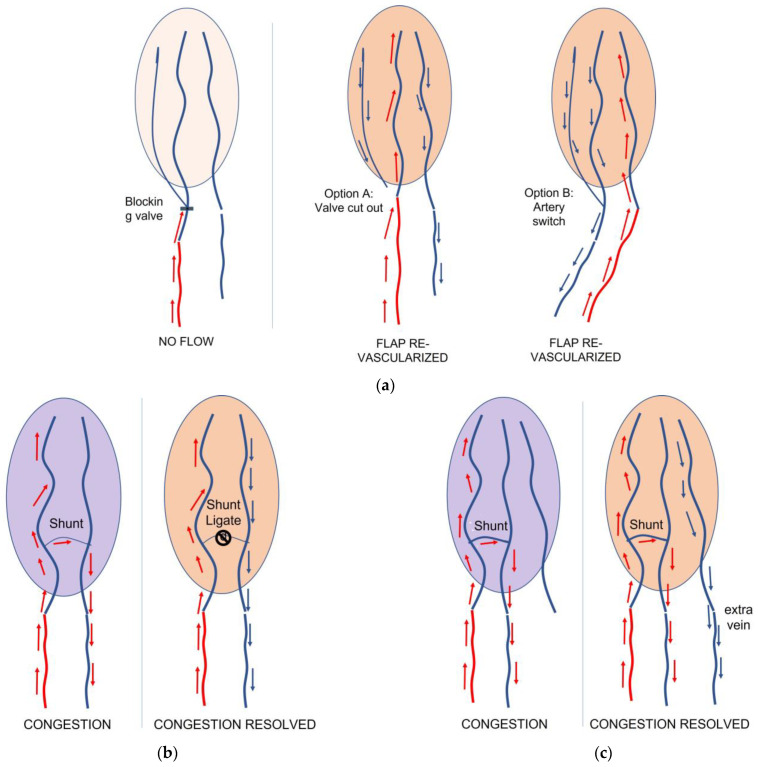
(**a**) In case of insufficient revascularization of the flap, a blocking valve before the flap can be the cause; accordingly, we suggest performing a more distal anastomosis to the same or to a different vein. Otherwise, we suggest switching the arterialized vein with the draining one as an alternative. (**b**) In case of congestion due to an Artero-Venous (AV) shunt, ligature is recommended; (**c**) alternatively, a second draining vein can be anastomosed in order to resolve the congestion.

**Figure 4 medicina-58-01065-f004:**
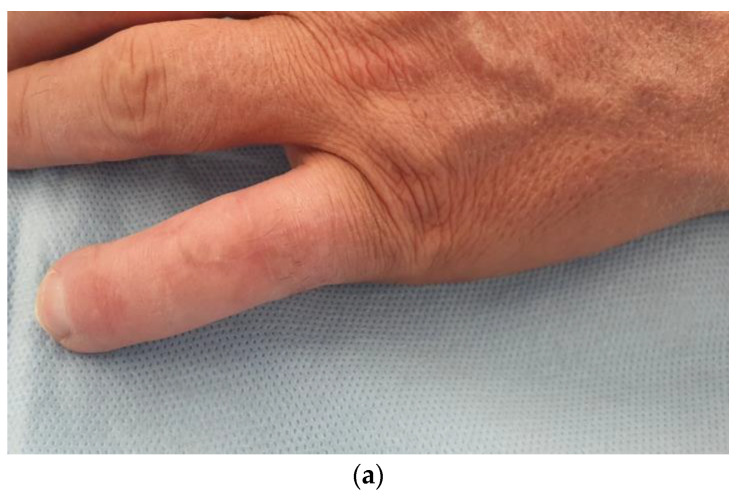
(**a**) 51 years old male patient with retracted skin, extensor tendon adhesions, and proximal interphalangeal joint post traumatic arthritis to the dorsum of the non-dominant left little finger; (**b**,**c**) tenoarthrolysis, joint replacement with a silastic implant, and reinforcement of the extensor tendon—the passive possible maximum flexion is shown; (**d**) harvesting of a retrograde free venous flap from the same forearm; (**e**) reconstruction of the skin of the dorsum of the finger—the arterialized vein is anastomosed end to end to the digital ulnar artery of the ring finger, dissected, and moved to the little finger; (**f**–**h**) result at 6 months with good finger extension and good flexion.

**Table 1 medicina-58-01065-t001:** Population of our series, with age, mechanism of the soft tissue defect, size of the defect, and concomitant important structures missing.

Case Number	Age	Sex	Mechanism	Side, Area to Be Covered	Defect Size cm^2^	Concomitant Defects
1	40	Male	Crush-avulsion	Dorsum left hand	102	Extensor tendons dig 2 to 4
2	19	Male	Infection	Left middle finger Dorsal and volar aspect	60	Ulnar digital nerve
3	47	Male	Burr Injury	Left little finger Dorsal aspect	10	
4	60	Male	Infection	Left Hand Dorsal aspect	40	
5	23	Male	Severe scarring after infection	Left hand 1st web space	40	1st web branches of the radial nerve
6	36	Male	Post traumatic stiffness	Right hand Dorsal aspect index finger	10	Proximal interphalangeal joint severe arthritis
7	32	Male	Scar contracture	dorsum left index	5	Radial digital nerve
8	64	Female	Infection after cat bite	Left index, dorsal aspect	9	
9	48	Male	Degloving injury	Left little finger, circumferential	84	
10	41	Female	Open ulna fracture	Left ulna head	50	Ulnar artery
11	52	Male	Infection	Right ring finger, dorsal defect	8	
12	35	Female	Crush injury	Right hand, dorsal defect to the ring and little finger	18	Extensor tendon defect to the little finger withBone defect at the PIP joint
13	45	Female	Avulsion / Friction burn	Left hand, dorsal aspect 2nd and 3rd ray	90	Extensor tendon to the index finger
14	23	Male	Burr injury	Left little finger, all the palmar aspect	21	
15	17	Male	Explosion	Left middle finger, radial side	8	Radial digital nerve of the index finger
16	29	Male	Crush Injury	Right index finger dorsal and volar aspect	15	Extensor tendon
17	40	Male	Scar contracture after chemical Burn	Left hand dorsal aspect	30	
18	27	Male	Post traumatic scar contracture	Left finger, dorsal aspect	14	
19	48	Male	Pseudoarthrosis with unstable scar	Right thumb, dorsal aspect	15	Bone defect of the proximal phalanx of the thumb
20	51	Male	Post traumatic stiffness	Left little finger, dorsal aspect	12	
21	52	Female	Previous first ray amputation	Right thumb dorsal and vola aspect	105	Bone defect
22	53	Female	Infection	Left hand dorsal aspect	63	
23	44	Male	Post traumatic skin contracture	Left hand, 1st web space	45	1st web branches of the radial nerve
24	42	Male	Thumb subamputation	Left thumb, dorsal aspect	9	Extensor pollicis longus
25	32	Male	Post traumatic stiffness	Right index, dorsal aspect	10	
26	37	Male	Burr injury	Left middle finger, radial aspect	6	
27	47	Male	Post traumatic stiffness and multiple operations	Left hand, 5th ray, dorsal aspect	12	
28	47	Female	Crush injury	Left ring finger, palmar aspect	6	
29	67	Male	Crush injury	Left ankle and heel	76	
30	43	Male	Soft tissue tumor	Right hand, palm	12	
31	28	Male	Explosion	Right thumb, all palmar surface	21	Ulnar digital nerve
**average**	**41**	**7f, 26m**			**32**	

**Table 2 medicina-58-01065-t002:** Technical details of the flaps. * In all flaps with a visible refill under the minute, the time was considered as one minute.

Case Number	Flap Size (cm^2^)	Length of Arterialized Vein (cm)	Number of Draining Veins	Structures Included Other Than Skin	Harvesting Time (min)	Time to Visible Refill (min) *	% Survival	Early Complications	Donor Site Closure
1	105	5	1	Palmaris longus Flexor carpi radialis	102	4	100%	acute ischemia: switch arter. vein	Partial Skin graft
2	77	3	2	Lateral cutaneous nerve of the forearm	60	7	100%		Direct
3	15	2	1	Palmaris longus	10	1	100%		Direct
4	55	3	1		40	4	100%	venous congestion: a-v shunt ligated	Direct
5	50	2	2	Medial cutaneous nerve of the forearm	40	10	100%		Direct
6	14	2	1		10	1	100%	intra-operative a-v shunt: immediate ligation	Direct
7	6	1.5	1	Peroneal nerve branch from the foot	5	1	100%		Direct
8	14	2	1		9	1	100%		Direct
9	110	3	2		84	1	100%		Skin graft
10	55	2	1		50	15	100%		Direct
11	12	1.5	1		8	4	100%	draining vein thrombosis: new anastomosis	Direct
12	24	3	1		18	1	100%		Direct
13	100	3	1	Palmaris longus	90	9	100%		Skin graft
14	28	2	1		21	30	100%		Direct
15	12	2	1	Lateral cutaneous nerve of the forearm	8	12	100%		Direct
16	18	2	1	Palmaris longus	15	1	100%		Direct
17	40	3	1		30	1	100%		Direct
18	18	2	1		14	1	100%	blocking valve: excision and new arterial anastomosis	Direct
19	20	3	2		15	1	100%		Direct
20	15	2	1		12	1	100%		Direct
21	136	3	2		105	25	85%		Skin graft
22	70	4	2		63	12	100%		Direct
23	55	3	1	Lateral cutaneous nerve of the forearm	45	5	100%		Direct
24	11	2	1	Extensor longus for the 4th toe	9	1	100%		Direct
25	12	2	1		10	1	100%		Direct
26	6	1	1		6	20	100%		Direct
27	18	2	1		12	1	100%		Direct
28	7	2	1		6	1	0%	venous congestion: draining vein new anastomosis	Direct
29	80	3	2		76	60	100%		Direct
30	15	2	1	Palmaris longus	12	1	0%	venous congestion: draining vein new anastomosis	Direct
31	28	3	1	Lateral cutaneous nerve of the forearm	21	1	95%		Direct
**average**	**40**	**2.5**			**32**	**7.5**	92%		

**Table 3 medicina-58-01065-t003:** Late complications and further procedures.

Case Number	Late Flap Correction	Late A-V Fistula	Secondary Donor Site Correction	Static 2 Points Discrimination	Secondary Functional Procedures	End Function
1					Extensor tenolysis	Full
2				7 mm	Extensor tenolysis	MCP 0–80° PIP 15–50° DIP rigid
3					Extensor tenolysis	MCP 0–90° PIP 5–90° DIP 0–40°
4					None	Full
5				8 mm	None	Full 1st web opening
6					Late PIP Fusion	MCP 0–90° DIP 0–30°
7				10 mm	none	Full No pain
8					Extensor and flexor tenolysis	MCP 0–90 PIP 30–70 DIP 0–10
9	Yes				Shortening of the little finger and neurotomy of digital nerves	MCP 0–80°
10						
11					None	Full
12	Yes				Finger separation	D5: MCP 0–90, PIP 10–70°, DIP Rigid. D4: MCP 0–90, PIP 60–100, DIP 10–30
13					Extensor tenolysis	D2: MCP 25–50°. PIP 0–50°, DIP 0–60°
14					None	Full
15	Yes	Yes	Yes	7 mm	None	Full
16	Yes				None	Full
17					None	Full
18					None	MCP 0–90°, PIP 30–80°, DIP 0–10
19					None	Kapandji 8 K-pinch 7 kg MCP 0–30. IP rigid
20					None	MCP 0–90° PIP 30–90° DIP 0–40°
21					New local flap for the tip of the thumb	Kapandji 7 K-Pinch 6 Kg
22		Yes			None	Full
23				7 mm	None	Full 1st web opening
24					None	Kapandji 7 Full 1st web opening
25					None	MCP 0–90° PIP 20–70° DIP 0–15°
26					None	Full
27					None	MCP 0–80° PIP 0–100° DIP 0° 50°
28					Flap necrosis: new flap	-
29					None	Full
30					Flap necrosis: new flap	-
31				7 mm	None	Kapandjy 8 MCP 0–40° IP 0–40° K-pinch 6 Kg

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
