# Peer review of "Retrograde Free Venous Flaps for Extremity Reconstruction: A Roadmap"

_medicina, 2022, doi:10.3390/medicina58081065_

Round 1

Reviewer 1 Report

1) Brief summary This is a case series of retrograde free venous flaps. The study included 31 cases. The manuscript is of clinical significance potentially worth publication, but needs recisions before considering publication.   2) The findings are The traditional venous flap concept is that the feeding recipient artery is connected with the flap vein in an antegrade manner. But, sometimes it is not possible for the different flap donor site and vessel choice. The authors showed it also could be a useful tool in some cases and the ICG vessel flow in Fig 1 showed how it works.   3) Its strength In the ordinary work of free flap transfer, we sometimes find that patients have very weak and tiny flap arterial pedicle system and it is difficult to do anastomosis. In that case, this method could be a very strong alternative, we can switch to a different way, connect it to a vein!   Vein is not one, many. Although it not perfect, it is worth doing and better than total failure. This article is a good reference.   4) Minor issue The minor issue to be addressed is the AV shunts. If the retrograde blood flow get through with the draining vein as a AV shunt, It becomes a problem. Ligature needed. But if with fine arterioles as a VA shunt form, I think it should be preserved. I wonder it is easily perceptible or distinguishable in the operation field.

Author Response

1) Brief summary This is a case series of retrograde free venous flaps. The study included 31 cases. The manuscript is of clinical significance potentially worth publication, but needs recisions before considering publication.

Thank you. We are keen to revise following suggestions  

2) The findings are The traditional venous flap concept is that the feeding recipient artery is connected with the flap vein in an antegrade manner. But, sometimes it is not possible for the different flap donor site and vessel choice. The authors showed it also could be a useful tool in some cases and the ICG vessel flow in Fig 1 showed how it works.  

Thank you. True but in reality the authors routinely use the retrograde form of the arterialized free venous flaps, as the phisiology seems more favourable, and, more cimportant, the 2 vessels (input vein and output vein) are on the same side of the flap. We slightly changed the text to reinforce the concept.

3) Its strength In the ordinary work of free flap transfer, we sometimes find that patients have very weak and tiny flap arterial pedicle system and it is difficult to do anastomosis. In that case, this method could be a very strong alternative, we can switch to a different way, connect it to a vein!   Vein is not one, many. Although it not perfect, it is worth doing and better than total failure. This article is a good reference.  

Thank you.

4) Minor issue The minor issue to be addressed is the AV shunts. If the retrograde blood flow get through with the draining vein as a AV shunt, It becomes a problem. Ligature needed. But if with fine arterioles as a VA shunt form, I think it should be preserved. I wonder it is easily perceptible or distinguishable in the operation field.

Thank you. This is a very fine comment and spot to the point. Intraoperatively it might be hard to distinguish between fine connecting vessels and a proper (unwanted) shunt. These are some clues: The draining vein is pulsating in synchrone manner with the arterialized vein and the same strength. The flap is intraoperatively very fast with a visible refill under a second. By flipping the flap upside down with the vessels connected, it easy under magnification to spot the A-V shunt. These three elements were the indications for immediate re exploration and AV shunt ligature or for a second vein anastomosis. We slightly modified the text accordingly

Reviewer 2 Report

The authors perform a retrospective analysis of 31 patients who underwent reconstruction using retrograde free venous flaps. They described detailed data of each case. I have some comments and queries.

1. Although it appears that these flaps are also used in cases where pedicled flaps, pedicled adipo-fascial flaps, and conventional free flaps including free perforator flaps, are applicable to limb reconstruction, the authors should clearly indicate whether they are used as the first choice or not, and in addition you should discuss it.

2. In Methods, inclusion or exclusion criteria should be mentioned if there were any.

3. In Surgical technique, you should not give details of each case. There seems to be a mixture of methods and results.

4. The point which authors would like to emphasize is that even large sized flaps can be used safely. On the other hand, the outcome of the donor site is important. In particular, the outcome of the donor site that required skin grafting must be documented. What about free flap donor site morbidity?

5. The conclusive statement should be a taking home message and could be more precise to justify the findings from the study.

Before acceptance, these major revisions should be addressed.

Author Response

The authors perform a retrospective analysis of 31 patients who underwent reconstruction using retrograde free venous flaps. They described detailed data of each case. I have some comments and queries.

1. Although it appears that these flaps are also used in cases where pedicled flaps, pedicled adipo-fascial flaps, and conventional free flaps including free perforator flaps, are applicable to limb reconstruction, the authors should clearly indicate whether they are used as the first choice or not, and in addition you should discuss it.

Thank you. We beg to disagree with the reviewer about going into a deep discussion about indications on which flap for what defect. The subject or the question  "what other flap could have been done for that defect"  is a source of endless debates in literature and all meetings involving flapping. All these discussion are in authors view, sterile discussions as any surgeons have his work horses and his preferences.  In the paper we clearly indicate that free venous flaps are in many cases our first choice ("We demonstrate how these flaps can be routinely used in elective situations; and report that the survival of these flaps is much larger than was previously believed. In reporting our results, we highlight and analyse our cases with their complications in order to contribute to the further understanding of these flaps.")

We also highlited how inthese era there is a hype for the SCIP flap ( the authors use also very often the scip flap) and we discussed the differences with the FVF it in the discussions. We are very sure that for every defect we have presented, there could have been a different solution. It is not the aim of the paper to prove that retrograde free venous flaps are a golden standard, and this will be a personal decision that any surgeon can take. True is that in many cases the FVF has been our first choice for the advantages described in the papaer i.e namely: FVF are  proximity free flap, the readyness of the flap, The ease of harvesting, the possibility of include tendons and nerves, the possibility to include very pliable and thin skin. We assumed that was implicit. We did not modify the text, but we are keen to modify it if required by the Editor.

  1. In Methods, inclusion or exclusion criteria should be mentioned if there were any. Thank you. Correct. we modified the text accordingly
  1. In Surgical technique, you should not give details of each case. There seems to be a mixture of methods and results. Thank you. We beg to disagree. As every case was different, as often in reconstructive surgery, we felt it was important to give details about most of the cases, as we really would have liked to give the reader most of the technical info about the procedures. This was done in the optic of offering a comprehensive report of our experience. Therefore methods are somehow complex and results are a mixture of data as the anatomical reconstructed parts were different. We highlighted anyway the important results as survival rate and revision rate in the text. Anyhow, If the Editor will request us to modify the text, we will be happy to comply.

  2. The point which authors would like to emphasize is that even large sized flaps can be used safely. On the other hand, the outcome of the donor site is important. In particular, the outcome of the donor site that required skin grafting must be documented. What about free flap donor site morbidity? Thank you. Correct. The forearm can be a challenging donor site for large flaps, especially reguarding aesthetic results.  We added in results the paragraph "donor site morbidity" with the results about donor sites.

  3. The conclusive statement should be a taking home message and could be more precise to justify the findings from the study. Thank you. we modified the conclusion for clarity

Round 2

Reviewer 2 Report

The authors perform a retrospective analysis of 31 patients who underwent reconstruction using retrograde free venous flaps. They described detailed data of each case.

Thank you for your revision. Much better than the last version. I think that the donor site morbidity section's addition made it much easier for readers to understand the usefulness and safety of this flap. Also, the conclusion became clearer.